# PTSBench: A Comprehensive Post-Training Sparsity Benchmark Towards Algorithms and Models

## ABSTRACT

With the increased attention to model efficiency, model sparsity technologies have developed rapidly in recent years, among which post-training sparsity (PTS) has become more and more prevalent because of its effectiveness and efficiency. However, there remain questions on better fine-grained PTS algorithms and the sparsification ability of models, which hinders the further development of this area. Therefore, a benchmark to comprehensively investigate the issues above is urgently needed. In this paper, we propose the first comprehensive post-training sparsity benchmark called PTS-Bench towards PTS algorithms and models. We benchmark 10+ PTS general-pluggable fine-grained algorithms on 3 typical computer vision tasks using over 40 off-the-shelf model architectures. Through extensive experiments and analyses, we obtain valuable conclusions and provide several insights from both PTS fine-grained algorithms and model aspects, which can comprehensively address the aforementioned questions. Our PTSBench can provide (1) in-depth and comprehensive evaluations for the sparsification abilities of models, (2) new observations for a better understanding of the PTS method toward algorithms and models, and (3) an upcoming well-structured and easy-integrate open-source framework for model sparsification ability evaluation. We hope this work will provide illuminating conclusions and advice for future studies of post-training sparsity methods and sparsification-friendly model design.

## CCS CONCEPTS

• **Computing methodologies** → **Neural networks**; *Object detection*; Object identification; • **General and reference** → **Evaluation**; **Computing standards, RFCs and guidelines**.

## KEYWORDS

Computer Vision, Model Compression, Post-Training Sparsification, Benchmark

## 1 INTRODUCTION

Although deep learning has been widely used in various fields, it requires a considerable amount of memory and computational power. To address this issue, many strategies have emerged to compress the model, including model quantization [10, 23–25, 39], model sparsification [8, 14, 16, 28, 29, 48], network distillation [12, 19], lightweight network design [45] and weight matrix decomposition [4]. One of

*ACM MM, 2024, Melbourne, Australia*

© 2024 Copyright held by the owner/author(s). Publication rights licensed to ACM.
ACM ISBN 978-x-xxxx-xxxx-x/YY/MM
https://doi.org/10.1145/nnnnnnn.nnnnnnn

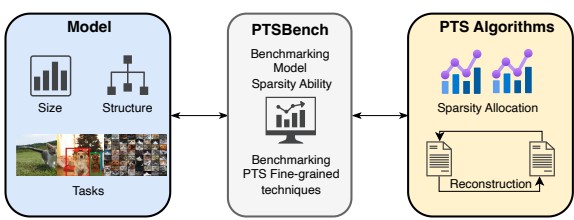

**Figure 1: The placement of PTSBench, which connects the models and PTS algorithms.**

the most representative methods is model sparsification, which involves removing unimportant weights from the model. Among all the sparsification methods, post-training sparsity (PTS) has received much attention in recent years because of its small training cost.

In the scenario of post-training sparsity (PTS), we are given a pre-trained dense model along with a small amount of unlabeled calibration data. We aim to generate an accurate sparse model without an end-to-end retraining process. Under these settings, several representative methods have been proposed, including POT [28], AdaPrune [22], and OBC [8]. These state-of-the-art (SOTA) methods have achieved almost no performance loss after sparsification. However, even though the high-performance PTS methods have reached, there are still two problems remain:

***Problem-1:*** **Post-training sparsity algorithm exploration is incomplete.** Current PTS methods [8, 22, 28] share the same sparsification paradigm: they first allocate sparsity rate to each layer to sparsify the model and then reconstruct the activation to recover the performance further. However, while current research follows this pipeline, it still lacks a fine-grained exploration of PTS techniques. For example, most PTS methods adopt layer-wise reconstruction granularity in the reconstruction process. However, block-wise reconstruction granularity has been proven effective in quantization approaches [30, 49] but is not explored in PTS algorithms. The absence of in-depth analysis of fine-grained techniques hinders further development of PTS approaches. Thus, benchmarking toward fine-grained techniques in PTS algorithms is urgently required.

***Problem-2:*** **Relationship between model and sparsity remains unexplored.** Current PTS research heuristically chooses commonly used models (e.g., ResNet and RegNetX) or datasets to validate their methods. However, it still lacks a comprehensive evaluation of the relationship between models and sparsity. In real-world applications, we often face the scenario that multiple network architectures with similar sizes can be used as the backbone, and we need to sparsify them before deployment for efficient inference. It is still an open question about which network architecture is

sparsity-friendly for better performance. Moreover, as the deployment platform varies, we also need to use different model sizes (e.g., different layer numbers). It is unclear whether a network architecture with different sizes is robust for the sparsity algorithms. In addition, model structures typically require various designs and modifications tailored to each task under different tasks. The extent to which these task-oriented designs and modifications are conducive to sparsity is also unknown. Therefore, conducting a comprehensive evaluation from an architectural perspective is also necessary from both theoretical and practical aspects.

To address the problems mentioned above, in this paper, we present **PTSBench**, a **P**ost-**T**raining **S**parsification **Bench**mark to evaluate the PTS technique from both algorithm and model aspects comprehensively. Starting from the real-world model production requirements, we carefully design 5 tracks for comparison. With over 8000 A800 GPU hours consumption, we benchmark 40+ classical off-the-shelf models, 3 typical computer vision tasks, and 10+ easy-pluggable fine-grained techniques in post-training sparsity. Based on the evaluation, we provide in-depth analyses of PTS methods from both algorithm and model perspectives and offer useful insights and guidance on PTS method design and validation.

Overall, our contributions can be summarized as follows:

(1) **Comprehensive benchmark.** We construct **P**ost-**T**raining **S**parsification **Bench**mark (PTSBench), which is the **first** systematic benchmark to conduct a comprehensive evaluation of PTS methods. It provides a brand new perspective to benchmark both post-training sparsity fine-grained techniques and sparsified models.

(2) **In-depth analysis.** Based on extensive experiments, we uncover and summarize several useful insights and take-away conclusions, which can serve as a guidance for future PTS method design.

(3) **Upcoming open-source framework.** We will release our open-source benchmark codes repository. Research communities can easily use our platform to evaluate sparsity approaches. It can also serve as a well-organized codebase for future research of PTS algorithms.

## 2 BACKGROUND

### 2.1 Post-training Sparsity

Post-Training Sparsity (PTS) aims to sparsify a pre-trained neural network while preserving its accuracy on a specific task. All of the current PTS researches [8, 11, 22, 28] adopt two-step sparsity paradigm including sparsity allocation and reconstruction, as shown in Figure 2. In the sparsity allocation procedure, a specific sparsity rate is allocated to each layer following a predetermined metric, and weights at corresponding positions are zeroed based on sparsity criteria. The reconstruction process involves employing a series of techniques to recover model accuracy drop from sparsification. We will give a detailed introduction to these two parts.

### 2.2 Sparsity Allocation

Many previous works have shown that allocating a more reasonable sparsity for each layer can lead to a more effective sparsification outcome [5, 11, 13, 16, 36]. Currently, sparsity allocation methods can be categorized into the following three types.

(1) **Heuristic based strategy.** In this type, the sparsity ratio for each layer is predetermined manually, such as uniform sparsity[9, 38].

(2) **Criterion based strategy.** Weights across all layers are ranked according to a specific metric, and a certain percentage of the weights with lower scores are set as zero. The corresponding sparsity rates for each layer can be naturally obtained[5, 7, 13, 43].

(3) **Learning based strategy.** This strategy learns sparsity rates for each layer by optimizing a loss function to achieve optimal sparsity allocation [11, 27].

Although current work proposed multiple sparsity rate allocation strategies, these works lack evaluation on broader architectures, sizes, and tasks. Moreover, while there is a strong focus on the effectiveness of the methods, it still lacks in-depth analysis, such as why allocating a sparsity rate in a certain way can reach high performance. These two limitations in current research pose the question of better practice for PTS algorithms.

### 2.3 Reconstruction

After sparsity allocation, PTS methods will apply reconstruction to reconstruct the sparse activation for compensating the accuracy loss caused by sparsity. In this paper, we benchmark three fine-grained pluggable techniques that are often identified as influencing the effectiveness of sparsity in this process: error correction, reconstruction input, and reconstruction granularity.

*2.3.1 Error Correction.* Error correction is widely used in many post-training quantization (PTQ) [17, 39, 40] methods. It aims to align the weight distribution after compression with the original weight distribution. However, current PTS methods do not comprehensively and systematically evaluate this procedure. Specifically, the error correction procedure can be written as follows:

$$\hat{\mathbf{W}}_s = \lambda \mathbf{W}_s + E(\mathbf{W}_d) - E(\lambda \mathbf{W}_s),$$
$$\text{and } \hat{b_s} = b_d + E(f(\mathbf{W}_d, \mathbf{X}_d)) - E(f(\hat{\mathbf{W}}_s, \mathbf{X}_d)),$$
$$\text{where } \lambda = \frac{\sigma(\mathbf{W}_d)}{\sigma(\mathbf{W}_s) + \epsilon}. \tag{1}$$

$\hat{\mathbf{W}}_s$ and $\hat{b_s}$ are the weights and biases after the error correction operation, and $\mathbf{W}_s$ denote the weights of the sparse model before correction. $b_d$, $\mathbf{W}_d$, and $\mathbf{X}_d$ are biases, weights, and input activation in the dense model, respectively. $f(\mathbf{W}, \mathbf{X})$ represents the convolutional operation performed by the layer on inputs $\mathbf{X}$ with weights $\mathbf{W}$. $E$ and $\sigma$ are the mean and standard deviation operators, $\epsilon$ is a small constant. In this way, we can correct the error caused by the distribution shift of weights and biases.

*2.3.2 Reconstruction Input.* During the reconstruction procedure, we can either use the output of the previous reconstruction unit from the dense model as the input or opt for the output after the previous sparsified units. In current research, the choice of reconstruction inputs is not aligned, but we find that it greatly impacts the results. Since the absence of systematic investigation in previous work, we also benchmark this technique in our PTSBench.

*2.3.3 Reconstruction Granularity.* In addition to error correction and reconstruction input, we also benchmark the reconstruction

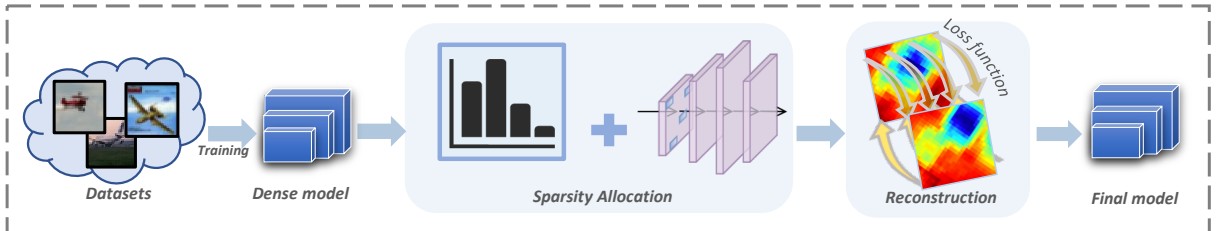

**Figure 2: The illustration of the overall Post-Training Sparsity pipeline, which is employed by most PTS methods.**

granularity in our PTSBench. Specifically, the reconstruction process can be conducted at different granularities. Many post-training quantization methods [30, 34, 49] prove that the reconstruction granularity has a large impact on quantization performance. However, the detailed impact of reconstruction granularities on a broad range of models is still unclear. Besides, the effectiveness of different reconstruction granularities still needs to be validated in PTS area. Therefore, inspired by [30], we mainly benchmark three reconstruction granularities:

(1) **Single reconstruction.** Reconstruct the weights based on each individual layer, which represents the smallest reconstruction granularity.
(2) **Layer-wise reconstruction.** Reconstruct the weights at the layer level. For instance, in a CNN, reconstruct in a CONV-BN-ReLU combination pattern.
(3) **Block-wise reconstruction.** Reconstruct the weights based on the block level (e.g., residual block).

In addition to the three aforementioned reconstruction granularities, some quantization methods also propose to use a net-wise reconstruction. However, although it is useful for quantization, we found this granularity will lead to poor performance because of overfitting. Hence, we do not include its performance for benchmarking in our PTSBench.

## 3 PTSBENCH: TRACKS AND METRICS

This section presents PTSBench, a benchmark for PTS methods from both algorithm and model aspects. Our evaluation consists of 5 tracks and corresponding metrics, as shown in Fig. 3, which provide comprehensive evaluation to address the limitations of current studies. All the metrics except $OM_{robust}$ are positive indicators. The model architectures we included are shown in Tab. 1.

### 3.1 Towards Fine-grained Algorithm

In our PTSBench, we benchmark "Sparsity Allocation" and "Reconstruction", which are two main procedures of PTS progress. To have a comprehensive evaluation, we conduct our experiments on 3 typical tasks: classification, detection, and image generation. For classification tasks, we test ResNet-18/50[15], RegNetX-200M/400M[42], MobileNetV2[45], ViT[3] (transformer-based model) under datasets ImageNet-1K [2]and ResNet32/56, VGG-19 [46] under datasets CIFAR-10/100 [26]. For detection task, we test RetinaNet-r18/50 [32]

on datasets MSCOCO-2017 [33] and MobileNetV1-SSD, MobileNetV2-SSDLite [35] on datesets PASCAL VOC07 [6]. For image generation task, we evaluate Stable Diffusion [44] on datasets LSUN-Churches/Bedroom [50].

① **Sparsity Allocation.** In order to evaluate the performance of different sparsity allocation methods and deeply investigate the characteristics and essence of a decent sparsity allocation, we choose 4 sparsity allocation approaches. Uniform sparsity allocation is a widely adopted human heuristic based method that allocates the same sparsity for each layer. L2Norm [13] and ERK [5]are both mask criterion based methods, while the latter is more meticulously designed. We choose these two methods for their effectiveness and high citations. We also include FCPTS [11] as a typical method that stands for methods based on learning, for it is the only one.

To better quantify the performance, we use the accuracy of dense models as a baseline and calculate the mean relative accuracy for all architectures and datasets on each task. Inspired by previous work [1], we define our overall metric (OM) by calculating the quadratic mean of the relative accuracies across 3 tasks as follows:

$$OM_{alloc} = \sqrt{\frac{1}{3}\left(\mathbb{E}^2\left(\frac{\mathbf{A}_{CLS}^s}{\mathbf{A}_{CLS}}\right) + \mathbb{E}^2\left(\frac{\mathbf{A}_{DET}^s}{\mathbf{A}_{DET}}\right) + \mathbb{E}^2\left(\frac{\mathbf{A}_{GEN}}{\mathbf{A}_{GEN}^s}\right)\right)}, \quad (2)$$

where $\mathbf{A}_*$ and $\mathbf{A}_*^s$ denotes the results obtained by calculating the metric of different tasks (i.e., accuracy for CLS, mAP for DET and FID [18] for GEN) of the dense and sparse models under different sparsity rate on a specific task (i.e., classification, detection and image generation), and $\mathbb{E}(\cdot)$ is the mean operator. Note that the Frechet Inception Distance (FID) is an indicator where lower values are better with a $[1, +\infty]$ value domain. We take the reciprocal to ensure the overall metric is in $[0, 1]$ interval. We sparsify the models we choose under different sparsity allocation algorithms and calculate the $OM_{alloc}$ for each algorithm.

To directly observe the impact of different sparsity allocation strategies, in this track, we measure the accuracy of the model immediately after sparsification without performing reconstruction.

The quadratic mean form is consistently employed across PTS-Bench to unify different tracks' overall metrics. This approach mitigates the undue influence of particularly poor performers on the metric, enabling a more precise evaluation of the comprehensive performance on each track.

② **Reconstruction.** In Section 2.3, we introduce 3 fine-grained reconstruction techniques that may have a considerable impact on

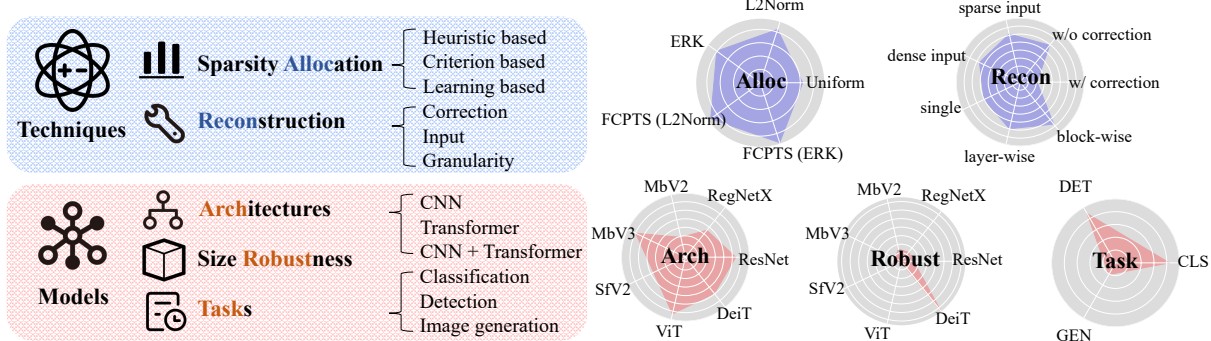

**Figure 3: Evaluation tracks of PTSBench. We benchmark the performance of PTS fine-grained algorithms and model sparsification abilities on a range of comprehensive evaluation tracks, including: "Sparsity Allocation", "Reconstruction", "Neural Architectures", "Model Size Robustness", and "Different Tasks". We illustrate an overview of the results of each track respectively on the right of the figure.**

**Table 1: Architecture repository.**

| Task | Arch. Family | Archs |
|------|-------------|-------|
| CLS | ResNet [15] | ResNet-18, ResNet-32, ResNet-34, ResNet-50, ResNet-56, ResNet-101, ResNet-152 |
| | RegNetX [42] | RegNetX-200M, RegNetX-400M, RegNetX-600M, RegNetX-800M, RegNetX-1600M, RegNetX-3200M, RegNetX-4000M, RegNetX-6400M |
| | MobileNetV2 [45] | MobileNetV2-x0.5, MobileNetV2-x0.75, MobileNetV2-x1.0, MobileNetV2-x1.4 |
| | MobileNetV3 [20] | MobileNetV3-x0.35, MobileNetV3-x0.5, MobileNetV3-x0.75, MobileNetV3-x1.0, MobileNetV3-x1.4 |
| | ShuffleNetV2 [37] | ShuffleNetV2-x0.5, ShuffleNetV2-x1.0, ShuffleNetV2-x1.5, ShuffleNetV2-x2.0 |
| | VGG [46] | VGG-19 |
| | ViT [3] | ViT-B/16, ViT-B/32, ViT-L/16 |
| | DeiT [47] | DeiT-Ti, DeiT-S, DeiT-B |
| DET | RetinaNet [32] | RetinaNet-R18, RetinaNet-R50 |
| | SSD [35] | MobileNetV1 SSD, MobileNetV2 SSD-Lite |
| GEN | Stable Diffusion [44] | Stable Diffusion V2 |

the effect while attracting little in-depth research. Therefore, we investigate them for detailed analyses in this track.

In this track, we compare the performance of the reconstruction procedure equipped with different proposed fine-grained techniques to the one without reconstruction:

$$\mathbf{R}_*^s = \mathbf{A}_*^{s,r} - \mathbf{A}_*^s, \tag{3}$$

where $\mathbf{A}_*^{s,r}$ denotes the results under different sparsity rates on all architectures and datasets of a task with decorated reconstruction. As the FID score for the generation task has a different scale compared with the other two tasks, we take the exponential for the results of generation tasks. For each fine-grained algorithms we benchmark, we compute an overall metric. The overall metric of this track can be calculated by:

$$OM_{recon} = \sqrt{\frac{1}{3}\left(\mathbb{E}^2\left(\frac{\mathbf{R}_{CLS}^s}{\mathbf{A}_{CLS}}\right) + \mathbb{E}^2\left(\frac{\mathbf{R}_{DET}^s}{\mathbf{A}_{DET}}\right) + \mathbb{E}^2\left(exp\left(\frac{\mathbf{A}_{GEN}}{\mathbf{R}_{GEN}^s}\right)\right)\right)}, \tag{4}$$

## 3.2 Towards Model Sparsification Ability

We also benchmark the sparsification ability of models in our benchmark. Our evaluation includes 3 tracks: "Neural Architecture", "Model Size Robustness", and "Different Tasks".

③ **Neural Architecture.** Although existing PTS methods evaluated the effectiveness on a wide range of models, the sparsity ability of the model itself remains uncovered. Therefore, We evaluate various neural architectures, including mainstream CNN-based and Transformer-based, to assess the model architecture from the perspective of sparsification ability. Specifically, for CNN models, we test ResNet, RegNetX, MobileNetV2, MobileNetV3 [20], ShuffleNetV2 [37]. For Transformer models, PTSBench includes DeiT [47] and ViT. Detailed information can be seen in Table 1. All accuracies of models are measured on ImageNet-1K datasets.

We use the following metric to describe the sparsification potential of a specific neural architecture:

$$OM_{arch} = \sqrt{\frac{1}{C}\sum_{i=1}^{C}\mathbb{E}^2\left(\frac{\mathbf{A}_{arch_i}^s}{\mathbf{A}_{arch_i}}\right)}, \tag{5}$$

where $\mathbf{A}_{size_i}$ denotes the accuracies of a specific architecture under the same sparsity rate of different sizes, and $C$ denotes the number

of sparsity rates. In simple terms, this metric evaluates the mean performance of models across different sparsity rates and model sizes within the same architectural framework.

④ **Model Size Robustness.** In real-world application scenarios, the requirements and resources differ, resulting in the scale of deployed models varying. We hope that the method will exhibit consistent effectiveness across models of the same architecture but with varying depths and widths. In other words, if a model demonstrates superior sparsity performance only in versions with a specific amount of parameters but shows great instability in performance across other parameter sizes, we do not recognize it as a sparsification-friendly architecture design. So, we benchmark model size robustness in this track and design our metric to quantify it. We first compute the quadratic mean relative accuracy for a model architecture of a specific size as follows:

$$G_{size} = \sqrt{\frac{1}{C} \sum_{i=1}^{C} \mathbb{E}^2 \left( \frac{A_{size_i}^s}{A_{size_i}} \right)}, \qquad (6)$$

where $A_{size_i}$ denotes the accuracy of a specific neural architecture with a specific size under a sparsity rate. Then, we calculate the standard deviation value among different sizes of one architecture.

$$OM_{robust} = std\left(\mathbf{G}_{size}\right), \qquad (7)$$

where $std(\cdot)$ denotes the standard deviation operator. To avoid the reconstruction process being affected by different model sizes, the experiments in this track are conducted without the reconstruction.

⑤ **Application tasks.** In various computer vision tasks, models are often combined and augmented before being deployed in applications. For example, ResNet is used as the backbone in detection tasks, with a neck and head connected afterward. In image generation tasks, CNNs and Transformers are combined for use. Therefore, there is a need to test sparsification ability from the aspect of tasks.

To this end, we evaluate 3 typical tasks which are commonly used in our PTSBench: classification, detection, and image generation. Similar to the overall metric for the neural architecture track, we build the overall metric for this track:

$$OM_{task} = \sqrt{\frac{1}{N} \sum_{i=1}^{N} \mathbb{E}^2 \left( \frac{\mathbf{A}_{task_i}^s}{\mathbf{A}_{task_i}} \right)}, \qquad (8)$$

where $\mathbf{A}_{task_i}$ denotes the accuracies set of a task under different sparsity rates using different models.

## 4 IMPLEMENTATION DETAILS

PTSBench is implemented using PyTorch framework [41]. We follow the pipeline introduced in Section 2.1 to sparsify the dense model. In the reconstruction process, we use the SGD optimizer for optimization. The momentum is set as 0.9, and the learning rate is set as $1e-4$. We randomly select 1,024 images from the training datasets as our calibration datasets and calibrate for 20,000 epochs. The batch size is set as 64. In our implementation, we observe that when the sparsity rate is lower than 50%, the performance drop after sparsification is negligible for almost all experiments. On the other hand, when the sparsity rate is higher than 80%, almost all setups undergo a collapse in accuracy. Therefore, we mainly present

the results under the {0.5, 0.6, 0.7, 0.8} sparsity rate. Results under more sparsity rates can be found in supplementary material.

## 5 PTSBENCH EVALUATION AND ANALYSIS

This section presents and analyzes the experimental results and evaluation conclusions in PTSBench. The results are shown in Tab. 2, Tab. 3, and Tab. 6. More details can be seen in the supplementary details.

### 5.1 Fine-grained Algorithm Tracks

As introduced in Sec. 2 and shown in Fig. 2, we benchmark the two significant procedures of PTS. The accuracy results of these tracks are shown in Tab. 2 and 3. The defined metric in Sec. 3.1 calculates the results.

*5.1.1 Sparsity Allocation: A Well-allocated Sparsity Results In High Performance.* We first present the evaluation results of different sparsity allocation strategies. To facilitate a more detailed analysis, we additionally report the root mean square components of the $OM_{alloc}$ for each task, denoted as MS, as well as its specific performance at each sparsity rate.

**The impact of sparsity allocation is crucial and significant.** Different sparsity allocation strategies vary greatly in results. Across various metrics, the Uniform strategy consistently shows the poorest performance, whereas learning-based methods uniformly exhibit good results. The gap between the two can reach up to 20%. ERK and L2Norm behave similarly, while FCPTS initiated with two strategies have an obvious difference, which implies that initialization matters a lot for learning-based methods and ERK possesses better potential for fine-tuning than L2norm (i.e., ERK sparsity allocation is closer to an optimal distribution).

**Effective sparsity rate allocation benefits from assigning lower sparsity rates to more sensitive layers.** We hope to further investigate the underlying reasons for the success of effective methods to better determine the sparsity rates for each layer. Therefore, we visualize the sparsity allocation using different methods. Fig. 4 shows the sparsity allocation of ResNet-32 on CIFAR-100 datasets. We can observe that effective methods unanimously allocate a lower sparsity rate to the final layer. This is because the last layer is directly related to the network's output features, making the output highly sensitive to changes in the weights of the last layer. Thus, the last layer is unsuitable for a large-scale sparsification.

We also observe that ERK and L2Norm commonly allocate a relatively low sparsity rate for the downsample layers, which implies that these methods consider downsample layers as sensitive layers. On the other hand, FCPTS tends to remove more weights from these layers while achieving better performance. This indicates that the poor performance is caused by mistakenly preserving more weights for sparsification-friendly layers.

*5.1.2 Reconstruction: By Making Simple Adjustments To The Pipeline, The Sparsity Effect Can Be Significantly Enhanced.* We report detailed results similar to track 1 in Tab. 3. Note that since the baselines vary, comparing the results of different techniques is not meaningful. We only compare technologies in the same aspect.

**Error Correction behaves differently in different tasks.** For classification tasks, we observe that error correction consistently

**Table 2: Benchmarking the sparsity allocation strategy of PTS methods. Blue: best in a column. Light blue: second best in a column. Red: worst in a column. Light red: second worst in a column.**

| Algorithms | CLS | | | | | DET | | | | | GEN | | | | | $OM_{alloc}$ |
|---|---|---|---|---|---|---|---|---|---|---|---|---|---|---|---|---|
| | 50 | 60 | 70 | 80 | MS | 50 | 60 | 70 | 80 | MS | 50 | 60 | 70 | 80 | MS | |
| Uniform | 96.94 | 84.52 | 65.67 | 32.55 | 72.88 | 98.31 | 93.75 | 80.89 | 47.77 | 80.94 | 16.40 | - | - | - | 16.40 | 63.59 |
| L2Norm | 98.22 | 95.68 | 86.17 | 52.53 | 84.05 | 98.66 | 96.92 | 94.08 | 65.86 | 88.97 | 78.47 | - | - | - | 78.47 | 83.94 |
| ERK | 97.67 | 94.34 | 83.62 | 56.13 | 84.00 | 99.23 | 98.08 | 95.14 | 76.40 | 92.30 | 62.61 | - | - | - | 62.61 | 80.61 |
| FCPTS (L2Norm) | 98.35 | 96.64 | 90.16 | 79.62 | 91.20 | 99.00 | 98.25 | 96.99 | 88.74 | 95.89 | 87.80 | - | - | - | 87.80 | 91.69 |
| FCPTS (ERK) | 97.51 | 97.85 | 93.79 | 88.57 | 94.55 | 99.19 | 98.58 | 97.16 | 91.95 | 96.76 | 91.71 | - | - | - | 91.71 | 94.36 |

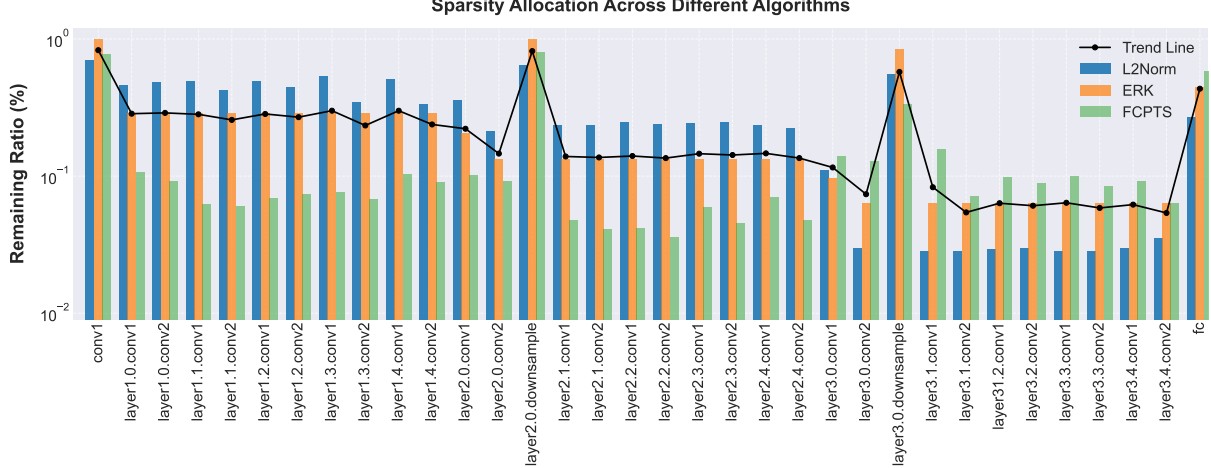

**Figure 4: Visualization of different sparsity allocation of ResNet-32 at a sparsity rate of 90% on CIFAR-100. The name of each layer is listed at the bottom. The black line denotes the average remaining ratio of the three algorithms. More visualization results can be seen in supplementary materials.**

**Table 3: Benchmarking the reconstruction techniques of PTS methods. Blue: best in a column. Red: worst in a column.**

| Algorithms | CLS | | | | | DET | | | | | GEN | | | | | $OM_{recon}$ |
|---|---|---|---|---|---|---|---|---|---|---|---|---|---|---|---|---|
| | 50 | 60 | 70 | 80 | MS | 50 | 60 | 70 | 80 | MS | 50 | 60 | 70 | 80 | MS | |
| w/ Correction | 11.87 | 35.06 | 59.32 | 32.90 | 36.68 | 0 | 8.71 | 1.83 | 0 | 3.73 | 88.32 | - | - | - | 88.32 | 55.26 |
| w/o Correction | 11.87 | 32.11 | 51.20 | 26.15 | 31.81 | 9.43 | 48.84 | 94.95 | 92.33 | 61.39 | 88.32 | - | - | - | 88.32 | 64.76 |
| Sparse Input | 11.85 | 34.71 | 59.07 | 33.86 | 35.87 | 9.28 | 48.69 | 94.79 | 92.02 | 61.20 | 88.32 | - | - | - | 88.32 | 65.40 |
| Dense Input | 11.30 | 33.91 | 56.44 | 23.40 | 32.84 | 8.73 | 47.46 | 92.40 | 88.40 | 59.25 | 88.06 | - | - | - | 88.06 | 64.14 |
| Singe | 9.40 | 33.26 | 59.73 | 42.75 | 38.01 | 3.83 | 41.72 | 83.46 | 71.89 | 50.24 | 88.11 | - | - | - | 88.11 | 62.54 |
| Layer-wise | 10.35 | 34.29 | 64.30 | 61.85 | 43.40 | 4.53 | 43.30 | 88.85 | 82.29 | 54.75 | 88.18 | - | - | - | 88.18 | 64.95 |
| Block-wise | 10.68 | 35.77 | 67.14 | 69.21 | 46.32 | 8.73 | 47.26 | 92.15 | 85.54 | 58.43 | 88.32 | - | - | - | 88.32 | 66.76 |

results in high performance under different sparsity rates, which is in alignment with experience in previous work[28]. However, the technique is performed diversely for detection and generation tasks. There is a significant collapse after applying error correction in detection tasks, while generation tasks seem to be insensitive toward the distortion of weights distribution. In object detection tasks, preserving the integrity of spatial information is crucial because detection involves not just "what" (identifying the object

categories) but also "where" (locating the positions of objects). Sparsification and subsequent adjustments to the weight distribution may disrupt the spatial features learned by the model, which is particularly critical for detection tasks. This could compromise the model's ability to accurately localize objects, affecting its overall detection performance.

**Use the output of sparse models as the input of reconstruction is beneficial.** From Tab. 3, we find that in most settings, sparse input can reach a higher performance increase than dense

**Table 4: Benchmarking sparsification potential of different tasks. Blue: best in a column. Red: worst in a column.**

| Tasks | Sparsity Rate (%) | | | | $OM_{task}$ |
|-------|------|------|------|------|-------------|
|       | 50   | 60   | 70   | 80   |             |
| CLS   | 98.24 | 95.68 | 86.17 | 52.53 | 84.05 |
| DET   | 98.66 | 96.92 | 94.08 | 65.86 | 88.97 |
| GEN   | 78.06 | 5.44 | 0.18 | 0 | 21.02 |

input, especially under a higher sparsity rate (e.g., for CLS tasks, 33.86 versus 23.40 under 80% sparsity rate). This may be because using the output from the sparse model can make the PTS algorithm aware of the reconstruction error from the previous layer, which avoids error accumulation across the network. Therefore, it is beneficial to use the output from the sparse model as the input for the reconstruction of the current unit.

**Block-wise reconstruction is always the best.** Blcok-wise reconstruction achieves the best results under most configurations, and layer-wise reconstruction outperforms the single reconstruction. For example, when sparsifying classification models at an 80% sparsity rate, block-wise reconstruction can outperform layer-wise by up to 3% and surpass single reconstruction by 16%. It is more important to carefully design the reconstruction granularity for classification tasks under a high sparsity rate. For instance, there are no significant differences at a 50% sparsity rate. However, at 80% sparsity rate, these differences become evident. However, this gap becomes apparent for detection tasks even at low sparsity rates.

Reconstructing on block-wise has the main advantage of potentially preserving the interactions between weights better. If the sparsity of a model is more concentrated at the block level, then reconstructing on a block basis might more effectively restore this sparsity. Moreover, block-level reconstruction could help reduce noise in the reconstruction process and provide more stable outcomes. On the other hand, reconstructing on layer-wise might lead to the loss of some crucial connections between weights across layers. This could limit the model's expressive ability and encounter difficulties in restoring sparsity. We also point out that reconstructing on a block-by-block basis can better preserve the locality of information and gradient flow compared to individual layers. This helps to maintain or restore the performance.

Reconstruction using a block-wise approach also has efficiency and resource consumption advantages. Given the large scale of feature maps, calculating loss can require significant time and memory resources. Reconstructing on a block basis can reduce the number of times loss calculations are needed. More details can be seen in the supplementary materials.

## 5.2 Model Tracks

We present the evaluation results of track 3 and track 4 in Tab. 6, and track 5 in Tab. 4. These results are calculated based on the metric defined in Sec. 3.2.

*5.2.1 Neural Architecture: Sparsity Potential Varies Across Different Architectures.* We report the evaluation results under different sparsity rates and overall metrics in Tab. 6.

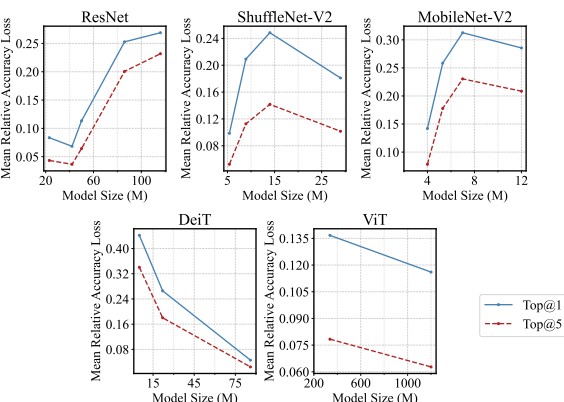

**Figure 5: The mean relative accuracy loss under different model sizes.**

**Models based on attention mechanisms possess greater sparsity potential.** From Tab. 6, we can find that ViT, MobileNetV3, and DeiT have their Overall Metrics (OM) positioned among the top across all models evaluated (top 1, 2, and 4, respectively). Compared to MobileNetV2, which possesses the worst $OM_{arch}$, MobileNetV3 significantly has its sparsity potential enhanced. The main difference between these two similar architectures is that MobileNetV3 introduced a Squeeze-and-Excitation Block (SE Block) [20], which is considered a lightweight attention mechanism[21]. The attention mechanism allows the network to focus more on features that are crucial for the final task, reducing dependency on less important features. Naturally, in scenarios of sparsification, it enables more effective preservation of information critical to performance.

**Training strategy can impact its sparsity potential.** DeiT has nearly the same architecture compared to ViT, while has obviously poorer performance. This is due to significant differences in their training strategies. ViT is pre-trained on extremely large datasets, such as JFT-300M, which likely enables it to learn more general feature representations. This makes it more robust against sparsification. While DeiT employs knowledge distillation as one of its core strategies, which aids in training efficient models with less data. However, if the sparsification method interferes with the features learned through knowledge distillation during the sparsification process, it could adversely affect DeiT's performance.

*5.2.2 Model Size Robustness: Different Model Architectures Tends To Vary In Model Size Robustness.* The results of evaluated models are shown in Tab. 6.

**A high sparsity potential for a model of a certain size ≠ a high sparsity potential for a model of all sizes.** Interestingly, we observe that DeiT and ResNet, which perform well on the $OM_{arch}$, exhibit poor performance on $OM_{robust}$, while ViT and MobileNetV3 behave consistently on both metrics. This suggests that the sparsity potential of a model and its size robustness don't exhibit a strict positive correlation. However, models with high sparsity potential are considered more likely to possess good model size robustness.

**A larger model size does not necessarily mean better sparsity ability.** We present the results of the mean relative accuracy loss (i.e., $G_{size}$ of Equation 6) of different model sizes in Fig. 5 to

**Table 5: The overall evaluation results of PTSBench. The results are listed from best to worst.**

| Track | | Results |
|---|---|---|
| Sparsity allocation | | FCPTS(ERK), FCPTS(L2Norm), L2Norm, ERK, Uniform |
| Reconstruction | Error correction | w/o correction, w/ correction |
| | Reconstruction Input | sparse input, dense input |
| | Reconstruction Granularity | block-wise, layer-wise, single |
| Neural architecture | | ViT, MobileNetV3, ResNet, DeiT, ShuffleNetV2, RegNetX, MobileNetV2 |
| Model size robustness | | ViT, MobileNetV3, MobileNetV2, ShuffleNetV2, RegNetX, RegNet, DeiT |
| Different tasks | | DET, CLS, GEN |

**Table 6: Benchmarking model sparsity potential and model size robustness. Blue and red: best and worst in a column. Light blue and light red: second best and second worst in a column. Note that $OM_{arch}$ is a positive indicator while $OM_{robust}$ is a negative indicator.**

| Models | Sparsity Rate (%) | | | | $OM_{arch}$ | $OM_{robust}$ |
|---|---|---|---|---|---|---|
| | 50 | 60 | 70 | 80 | | |
| ResNet | 99.38 | 97.86 | 91.38 | 48.36 | 86.81 | 0.286 |
| RegNetX | 97.98 | 94.57 | 83.65 | 45.17 | 83.04 | 0.270 |
| MobileNetV2 | 97.32 | 93.48 | 79.80 | 29.51 | 79.77 | 0.219 |
| MobileNetV3 | 97.93 | 95.54 | 89.14 | 65.32 | 87.94 | 0.172 |
| ShuffleNetV2 | 96.50 | 92.77 | 82.96 | 54.04 | 83.25 | 0.258 |
| ViT | 99.50 | 97.52 | 89.99 | 62.41 | 88.61 | 0.014 |
| DeiT | 98.08 | 95.53 | 85.81 | 58.25 | 85.88 | 1.485 |

deliver a further detailed analysis. From the figure, we can draw the following discoveries. (1) For CNNs, the mean relative accuracy loss first increases and then decreases as the model size increases. So, a middle-sized network is the sweet point for CNN-based network architectures. (2) For DeiT and ViT, the mean relative accuracy loss decreases as the model size increases, which shows Transformer is more amenable to sparsification under a larger model size.

One possible explanation for the divergence between CNNs and Transformers goes to the fundamental differences in architecture between them. The self-attention mechanism of Transformers provides a way to optimize and adapt to the effects of sparsification on a global scale. In contrast, the localized feature extraction and hierarchical dependencies of CNNs make them more sensitive to reductions in parameters, especially in larger models. Thus, Transformer models are better able to be sparsified as model size increases, whereas large CNN models may face performance challenges due to sparsification.

*5.2.3 Different Tasks: The PTS Method Needs Further Development In Generation Tasks.* In Tab. 4, we present the benchmark results of different tasks calculated by the metric designed in Sec. 3.2.

**In detection models, the PTS method demonstrates better performance compared to classification models.** From Tab. 4, we find that $OM_{task}$ can reach up to 88.97 in detection tasks, whereas classification task scores 84.05. This implies that attaching subsequent structures (such as the neck [31] and head parts in

detection models) to a backbone does not reduce sparsity potential; it can even make the model more sparsity-friendly due to the introduction of additional parameters.

**The PTS method still urgently requires further exploration in the field of image generation.** Generation tasks can only maintain precision at 50% sparsity rate, with a collapsing performance on higher sparsity rates. Sparsification methods used for other types of models may be unsuitable for Diffusion models, or at least they may perform poorly without proper adjustments. This could be because these methods do not consider the unique operating mechanisms of Diffusion models and the distribution of their parameters.

## 5.3 Overall Results

The overall evaluation results are shown in Tab. 5. Note that the algorithms or models that perform best in the table do not necessarily perform best under all experimental setups, but exhibit the best performance on multiple aspects.

## 6 CONCLUSION

In this paper, we systematically propose a **P**ost-**T**raining **S**parsification **Bench**mark called **PTSBench**, which is the first comprehensive benchmark towards the post-training sparsity (PTS). From an algorithm perspective, we benchmark 10+ PTS components on 3 computer vision tasks. From a model perspective, we benchmark 40+ network architectures. PTSBench aims to establish a comprehensive and in-depth analysis of PTS algorithms, providing useful technical guidance for future research. Our benchmark is highlighted by fertilizing the community by providing the following: (a) comprehensively evaluate models from the perspective of PTS. (b) new observations towards a better understanding of the PTS fine-grained algorithms. (c) an upcoming open-source platform for systematically evaluating the model sparsification ability and pluggable sparsification algorithms. We plan to explore a broader range of model architectures and tasks in future work. We hope our PTSBench can provide useful advice for future studies.

Our PTSBench also has limitations: (1) We benchmark PTS methods on three vision tasks, and it is better to include more tasks like natural language processing in our PTSBench. (2) The number of PTS algorithms available for study is relatively small in the current research. As the PTS community becomes more fertilized, it is desirable also to include more PTS algorithms in our PTSBench. Considering the aforementioned limitations, we will continue to include more methods and tasks in our PTSBench platform.

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
