# OpenReview forum: "PTSBench: A Comprehensive Post-Training Sparsity Benchmark Towards Algorithms and Models"
_acmmm.org/ACMMM/2024/Conference — MM2024 Poster_

### Official Review · Reviewer_6vtT · 2024-05-12

**Rating:** 3
**Confidence:** 3

**Summary:**

This paper proposes a post-training sparsification benchmark (PTSBench) to explore the performance effect of PTS methods. In particular, the authors evaluate the PTS technique from both algorithm and model aspects. For the algorithm, this paper mainly evaluates the performance effect on the sparsity allocation and reconstruction strategies. For the model, neural architecture, model size robustness and application tasks are comprehensively analyzed in PTSBench. Through experimental results, the authors evaluate the conclusions in PTSBench.

**Strengths:**

1. Clear motivation to construct PTSBench.

2. Well-written and easy to follow.

3. Comprehensive experiments to evaluate the performance effect on multiple factors of PTS methods.

**Limitations:**

1. This paper is like an experimental report of PTS methods, which does not have technical novelty.

2. There are many PTS methods, but the authors only discuss partial methods, which is not sufficient to support the experimental conclusion.

**Suitability:**

1

---

### Official Review · Reviewer_nMoz · 2024-05-31

**Rating:** 4
**Confidence:** 4

**Summary:**

The authors propose PTSBench, a comprehensive post-training sparsity (PTS) benchmark for evaluating the sparsification capabilities of neural network models and algorithms. It investigates ten PTS algorithms across over forty model architectures in three computer vision tasks, providing a systematic examination of post-training sparsity effects. The study identifies and addresses key research gaps, such as fine-grained algorithm analysis and model-sparsity relationships, offering a new benchmark tool and an upcoming open-source framework for future research.

**Strengths:**

PTSBench fills a significant gap by providing a structured and comprehensive means to evaluate post-training sparsity.

The research's breadth is good, examining a wide range of algorithms and model architectures across multiple tasks, which enhances the validity and applicability of the findings.

The upcoming open-source framework and detailed analysis are poised to be highly beneficial for both academic researchers and industry practitioners, facilitating easier adoption and experimentation.

**Limitations:**

While the benchmark covers several CV tasks, its application to other domains like NLP isn't addressed, which could limit its utility across broader AI fields.

The presentation can be improved. As this is a benchmark paper and they conduct many experiments, they should highlight their explored results in each subsection.

The formulation of this benchmark paper looks like another paper, BiBench (ICML 2023), especially about the overall metric formulation.

**Suitability:**

3

---

### Official Review · Reviewer_RjjS · 2024-06-01

**Rating:** 6
**Confidence:** 3

**Summary:**

This paper talks about a proposed post training sparsification (PTS) benchmark, studying three different aspects, including different sparsification strategies, different model architectures, and different tasks. It benchmarks 10+ PTS general-pluggable fine-grained algorithms on 3 typical computer vision tasks (classification, detection, and generation) using over 40 off-the-shelf model architectures. They deliver algorithm-wise, model-wise, and task-wise insights around PTS.

**Strengths:**

1. Clear writing and insightful figures. The paper is writing really well, detailing the branches of PTS, the current limitations of PTS evaluations, and showing the experimental results following the sequential order of algorithms, models, and tasks. To demonstrate the extensive experiments of existing PTS algorithms, different types of figures are properly used. For example, Figure 3 gives a clear picture of what the benchmark wants to study; Table 2 exhibits the performance differences of various PTS strategies across tasks; Figure 4 shows an example of how different strategies affect the final layer-wise sparisities, etc..
2. Insightful and interesting conclusions. After benchmarking different algorithms on different architectures, the work shows many interesting results. Especially, some of the interesting findings are concluded:
a. Block-wise reconstruction is always the best.
b. Models based on attention mechanisms possess greater sparsity potential.
c. A high sparsity potential for a model of a certain size ≠ a high sparsity potential for a model of all sizes.
3. The generation task study is not often seen in prior works. Most of previous studies discuss around the discriminative tasks like classification and object detection. The study of generation tasks by PTS is really valuable to the community and the urgent need is called for the further exploration.

**Limitations:**

1. Suggestion: time efficiency is one important metric for evaluating PTS. All the results are put in the suuplementary due to the space limitation, but show some representative table or figure in the main manuscript may better help the audience to have a picture of these PTS algorithm consumptions.
2. You mentioned [27] is also a learning-based PTS in L185, but why in L316, saying FCPTS is the only one.
3. I know it is not easy to conduct lots of GEN PTS, where only SDv2 is included in Table 1. Considering many CLS PTS are based on CNNs, is that possible to include some GAN-based GEN PTS as well? Will that affect your conclusions? Considering the time and efforts, you can also answer why not try some DDPM or DDIM based tasks which might be more fundamental compared to SD.

**Suitability:**

3

---

### Official Review · Reviewer_xKmZ · 2024-06-02

**Rating:** 4
**Confidence:** 3

**Summary:**

In this paper, the authors conducted a comprehensive analysis of post-training sparsity from the perspectives of algorithms and models. This analysis includes examining PTS fine-grained algorithms and model sparsification abilities on a range of comprehensive evaluation tracks, including: "Sparsity Allocation", "Reconstruction", "Neural Architectures", "Model Size Robustness", and "Different Tasks", offering significant insights for current researchers in the PTS field. Overall, the writing is good and easy to follow. Extensive experiments are conducted from the above five perspectives, and the conclusions are easy to draw from the provided results, providing a better understanding of the PTS method towards algorithms and models.

**Strengths:**

(1)The authors provide the first in-depth and comprehensive evaluations for the sparsification abilities of models by verifying 10+ PTS components on 3 computer vision tasks, consisting of 40+ network architectures.

(2)The final evaluation results of PTSBench can provide new observations towards a better understanding of the PTS fine-grained algorithms for future researchers.

(3)The thorough experimentation significantly enhances the value of the analysis, providing a solid foundation from which clear conclusions can be drawn.

**Limitations:**

(1)In the first problem of existing high -performance PTS methods, the authors claim that current research still lacks a fine-grained exploration of PTS techniques, where layer-wise reconstruction is adopted and the block-wise reconstruction granularity is ignored.  In lines 258 to lines 262, the layer-wise reconstruction denotes layer like CNN, and block-wise reconstruction represents block like residual block. However, in general, layer-wise reconstruction is more fine-grained because it focuses on reconstructing each layer of the model in detail. In contrast, block-wise reconstruction is more coarse-grained because it deals with reconstructing higher-level units composed of multiple layers. The authors should explain such misunderstanding clearly.

(2)In equation 5, the notations are inconsistent between $A_{arch}$ and $A_{size}$.

(3)A suggestion is that it would be better to directly explore and present the sparsity rate of different layer in figure 4.

(4)In line 845 and figure 5, the discovery of (1) can not be fully drawn from the provided figure5. It is clear that the mean relative accuracy loss of ResNet does not first increase and then decrease as the model size increase.

(5)Figure 1 is not cited in the main paper.

**Suitability:**

2

---

### Meta-Review · Area_Chair_zPJp · 2024-06-29

**Recommendation:** Accept (Poster)
**Confidence:** 5

**Metareview:**

This paper proposed the first comprehensive post-training sparsity benchmark called PTSBench towards PTS algorithms and models The paper received 4 acceptances from reviewers after rebuttal. AC has looked at reviews and agreed with the acceptance of this paper.